# OpenReview forum: "SEMU: Singular Value Decomposition for Efficient Machine Unlearning"
_ICML.cc/2025/Conference — ICML 2025 poster_

### Official Review · Reviewer_YqtF · 2025-03-13

**Overall Recommendation:** 2

**Summary:**

The paper proposed a machine unlearning method that only fine-tunes by the subspace of the gradient orthogonal to the weight. It claims to effectively “unlearn” forgetting sets while eliminating the dependency on the original training dataset.

**Claims And Evidence:**

see Methods And Evaluation Criteria

**Essential References Not Discussed:**

The paper lacks discussion and comparison with some related methods [1–4]. In particular, references [1] and [2] are highly relevant but are not discussed in the paper.

**Experimental Designs Or Analyses:**

See Methods And Evaluation Criteria

**Methods And Evaluation Criteria:**

The paper does not compare its method with some related approaches [1–4].

There is a lack of a quality measure for removing the nudity concept.

While the paper points out that SalUn’s performance drops quickly with reduced data availability, the comparison to the proposed method is missing.

[1] Machine Unlearning via Null Space Calibration. IJCAI-2024

[2] SAP: Corrective Machine Unlearning with Scaled Activation Projection for Label Noise Robustness. AAAI2025

[3] Fast Machine Unlearning Without Retraining Through Selective Synaptic Dampening. AAAI 2024

[4] Deep Unlearning: Fast and Efficient Gradient-Free Class Forgetting. TMLR 2024

**Other Comments Or Suggestions:**

see weaknesses

**Other Strengths And Weaknesses:**

1) The main weakness is the missing discussion and comparison to related work [1–4], which limits the paper’s contribution.
2) The paper overclaims that no remaining dataset is needed, yet experiments still use it.
3) The performance of the proposed method lags behind existing methods.
4) There is insufficient discussion about cases where the gradient has no projection on the weight space.
5) It is unclear how the method would be applied to transformer architectures or convolution blocks.
6) A timing analysis would be beneficial to demonstrate the efficiency of the proposed method.

**Questions For Authors:**

The training process of the component “R” is not clearly explained, and the pseudo code does not show its training process.

What do the bold results in Table 4 represent? Why is only the best TA highlighted? The UA is an essential measure of machine unlearning.

**Relation To Broader Scientific Literature:**

The paper claims to eliminate the dependency on the original training dataset. However, experiments still use the remaining set to achieve better performance. Moreover, the proposed method appears to underperform compared to existing methods.

**Theoretical Claims:**

There is a lack of proof on how the proposed method works when the gradient is in the weight space. Table 1 indicates that the proposed method does not unlearn the forgetting sets.

---

> ### Author Rebuttal · Authors · 2025-03-31
>
> **Referencing other works** Thank you for highlighting these works. We will discuss the differences between SEMU and these methods and include this analysis in the camera-ready version of our work. Regarding [1], our method does not rely on samples or gradients from the remaining dataset, nor do we perform pseudo-labeling of the forget class to the most activated incorrect class for each unlearning sample. The work [2], published at AAAI25 after the ICML submission deadline, was not known to us at the time. We note that [2] operates on the representation of a trusted dataset, whereas SEMU focuses on gradients and does not require any additional datasets. In the case of [3], multiple datasets (forgetting and remaining) are used, but the authors employ Fisher and Hessian matrices to select important parameters, unlike SEMU, which uses SVD. Lastly, [4], similar to [2], performs unlearning by identifying important parameters in the representation space. It requires the identification of remaining and forget spaces using representations from both datasets. In summary, while various approaches use different projection methods, none operate without a remaining dataset. This discussion will be included in the revised version.
>
> **Quality metrics for nudity concept** We provide the additional qualitative and quantitative evaluation for this in Sections 2, 3, and 4 in https://anonymous.4open.science/r/icml2025_submission_3162/REBUTTAL.md .
> In particular, we use MSE and CLIP measures to compare SalUn and SEMU to Stable Diffusion on NSFW (Tab. 1) and safe (Tab. 2) prompts. Especially, comparing the visual samples in Section 4 shows the quality difference.
>
> Tab. 1
> |Method|CLIP(T,I)|CLIP(I,I)|MSE|
> |:-|:-:|:-:|:-:|
> |SD|0.285| - | - |
> |SalUn|0.131|0.529|0.101|
> |SEMU (OUR)|0.280|0.747|0.025|
>
> Tab. 2
> |Method|CLIP(T,I)|CLIP(I,I)|MSE|
> |:-|:-:|:-:|:-:|
> |SD|0.268| - | - |
> |SalUn|0.196|0.630|0.083|
> |SEMU (OUR)|0.267|0.855|0.023|
>
> We observe almost perfect sampling on safe prompts and have better MSE on NSFW prompts as well.
>
> **Remaining dataset usage** SEMU is not limited to operating only in scenarios without a remaining dataset. One of its notable properties is its ability to function effectively in both conditions, with and without remaining datasets. Furthermore, we present results using the remaining dataset (always indicated as SEMU_{remain}). These results also indicate that the remaining dataset has minimal impact on SEMU's performance.
>
> **Performance lag** We agree that there is a difference in performance. This difference is mostly caused because of a different approach, i.e., we want to have an unlearning method working reasonably well, even without access to any examples from the remaining dataset, which is really hard for the SalUn.
>
> Finally, we observed that SalUn's concept of unlearning leads to catastrophic behaviour on normal (safe) prompts and conceptually far on NSFW as well. We provide the quantitative and qualitative results in Sections 2, 3, and 4 in additional experiments (https://anonymous.4open.science/r/icml2025_submission_3162/REBUTTAL.md).
>
> **Table 1 concerns** When looking at Table 1, we notice that the unlearning accuracy (UA) is low, indicating that the model struggles to perform correctly on the forget dataset. This performance is comparable to other unlearning methods such as GA, IU, BS, BE, and FT. Regarding the results of the Membership Inference Attack (MIA), our findings show low recognition rates, suggesting that the attack does not identify the data as part of the model's training set. Based on these observations, we conclude that SEMU effectively performs the unlearning task.
>
> **Discussion on gradient projection** Please see ablations in rebuttal for Reviewer 8DF6.
>
> **Application to convolution**
> For a 2D convolutional layer, we treat each channel as a separate matrix (while restricting ourselves to the maximum 'sparse' value across all channels – see line 111 in the code [here](https://anonymous.4open.science/r/icml2025_submission_3162/SEMU/Classification/unlearn/own/transform_model.py)). Notice that we can flatten the kernel dimensions, and by performing matrix multiplication along the appropriate dimensions, we obtain a similar operation to a standard convolution.
>
> Then, similar to the linear layer case, we compute $U W V^T$, but here $W$ corresponds to the channel (one dimension represents the flattened kernel). Since the tensors $U, W, V^T$ have more than two dimensions, we use the `einsum` function for computation. After the multiplication, we reshape the dimensions where we previously flattened (for the kernel) and finally permute the tensor to match the correct weight format for the convolutional layer.
>
> All these operations are performed in just three lines of code (see lines 78-80 in [this file](https://anonymous.4open.science/r/icml2025_submission_3162/SEMU/Classification/unlearn/own/utils.py)).
>
> **Time analysis** Please see the response for Reviewer Qmc3.

---

### Official Review · Reviewer_tPuz · 2025-03-14

**Overall Recommendation:** 3

**Summary:**

This paper proposes a machine unlearning (MU) method named Singular Value Decomposition for Efficient Machine Unlearning (SEMU). The authors disentangle the gradients of parameter weights with Singular Value Decomposition (SVD) to identify the important proportion for MU. They keep all original weight matrices frozen and concatenate a processed SVD output of the projected accumulated gradient matrix on each of them. In particular, each accumulated gradient matrix is projected in a direction perpendicular to the existing weights before SVD, and all elements of the diagonal matrix in the SVD output are initialized as 0. During the unlearning training procedure, only the modified diagonal matrices are updated. The authors focus on two kinds of visual tasks, image classification, and generation, to validate their method. For the former, the authors conduct experiments on random data forgetting and class-wise forgetting. For the latter, class unlearning and concept unlearning are selected to evaluate their method.

## update after rebuttal
This reviewer appreciates the authors' efforts to address the raised concerns. After checking the authors' responses, the major concerns of this reviewer have been solved. Therefore, this reviewer choose to raise the rating.

**Claims And Evidence:**

- The authors state in the contributions that they propose a remaining dataset-free scenario for machine unlearning. However, some existing work has focused on this topic, such as [R1] and [R2].

- The authors try to disentangle a specific operator A with a projection $\rm\pmb{A = UU^TAV^TV}$, where $\rm\pmb{U}$ and $\rm\pmb{V}$ are orthogonal matrices. They combine it with concentrating the gradient information into a small proportion intuitively. This reviewer considers that intuitive thought is not strict enough, and an interpretation should be needed.

- The authors propose a projection operator $p_{A, B}(X)$ and claim that “This projection is particularly useful when applied to the gradient matrix G.” Why is this projection useful for the gradient matrix? They should present an explanation for it.

- The authors claim in the experiment part that SalUn is the most similar approach (compared with the proposed method). Why?

[R1] Bonato, Jacopo, Marco Cotogni, and Luigi Sabetta. "Is Retain Set All You Need in Machine Unlearning? Restoring Performance of Unlearned Models with Out-of-Distribution Images." European Conference on Computer Vision. Cham: Springer Nature Switzerland, 2024.

[R2] Cheng, Xinwen, et al. "Remaining-data-free Machine Unlearning by Suppressing Sample Contribution." arXiv preprint arXiv:2402.15109 (2024).

**Essential References Not Discussed:**

One characteristic of the proposed method is no need for remaining dataset-free. There is a published article [R1] exploring this topic. Additionally, this method is similar to LoRA [R3], which should be discussed further.

**Experimental Designs Or Analyses:**

Yes. This reviewer has checked all experiment settings and results, and there are some issues.

- The results are not satisfactory enough.
  - Image classification: The UA and MIA are commonly worse than those of SalUn.
  - Image generation: In Table 6 and Table 7, most results are worse than those of SalUn.

- Some experiment settings need to be clarified.
  - TA: How is the test set constructed? Is it the same as the original dataset or processed like the unlearned training set?
  - Forgetting data: How to replace the original data labels?

- Some settings do not align with the baselines.
  - Image classification
    - Missing datasets:
      - RL: Lacuna-10 and Lacuna-100
      - l1-sparse: ImageNet
      - BS and BE: Vggface2
      - SalUn: SVHN and TinyImageNet
    - Missing model:
      - SalUn: Swin-T
  - Image generation
    - Missing dataset
      - FMN: ConceptBench

- In the class unlearning of image generation, the authors only unlearn the "airplane" class from CIFAR-10. Additional experiments on unlearning other classes should be conducted to present the stability of the proposed method.

**Methods And Evaluation Criteria:**

- Methods
  - The novelty is limited.
    - Loss function: The loss functions are the same as that in SalUn.
    - Training method: The implementation of the method is similar to  low-rank adapters, such as LoRA [R3], which introduces extra parameters to the original model and only updates them during follow-up finetuning. The authors should discuss these methods and compare them with theirs.
- Evaluation Criteria
  - Some metrics are unclear.
    - MIA: How to compute?
    - UA, RA, and TA: The accuracy of the image classification task is easy to guess. However, how to measure the accuracy of image generation task?
  - An important metric is not discussed.
    - Train time: Although the authors show the rate of training parameters, the training time needed to achieve the best performance is a more straight metric to measure the computation efficiency.

[R3] Hu, Edward J., et al. "Lora: Low-rank adaptation of large language models." ICLR 1.2 (2022): 3.

**Other Comments Or Suggestions:**

- There exist some writing issues.
  - Typos in the caption of Figure 2
    - remain unaltered **adn** they are derived from
  - Wrong jump links in section 6, Image Generation
    - requiring only a small fraction of the trainable parameters (see **Fig.G.1 and Fig. G.1**)
    - the images in **Figure 6** show that after SEMU
  - Wrong equation writing in Eq. 17 and Eq. 18
    - A missing equator, or redundant $L_c(\theta_u)$ and $L_g(\theta_u)$
  - Unexplained marks in tables
    - up and down arrows in Table 6 and Table 7

- Some signs can be uniform.
  - In section *Truncated SVD*, $\rm\pmb{\Sigma_r}$ and $\rm\pmb{U_r}$ represent the orthogonal matrices of SVD output, while $\rm\pmb{A_r}$ and $\rm\pmb{B_r}$ do in section *Selecting most importan subspace of $\Sigma$*.

- Some items should be explained further.
  - In Eq. 18, the generation loss contains a mean squared error loss $ℓ_{MSE}(θ_u; D_r)$. What are the details of this loss?
  - In  Algorithm 2 and Algorithm 3, a description says, "When using retrain mode." What is the meaning of retrain mode?

**Other Strengths And Weaknesses:**

None.

**Questions For Authors:**

- Why do you consider that you propose a remaining dataset-free scenario?
- What are the innovations of your method when compared to low-rank adapters, like LoRA [R3]?

**Relation To Broader Scientific Literature:**

- The remaining dataset-free scenario has been explored in [R1] and [R2].
- The proposed method is similar to low-rank adapters, such as LoRA [R3].
- The proposed method do not surpass existing work like SalUn.

**Theoretical Claims:**

This reviewer has quickly checked all the mathematical proofs and found no errors in general.

---

> ### Author Rebuttal · Authors · 2025-03-31
>
> We appreciate your feedback. Below we address the concerns.
>
> **Previous works R1 and R2** Regarding R1, we would like to highlight that it requires an additional surrogate dataset $\mathcal{D}^{sur}$, which is not required for SEMU. This means our approach does not rely on extra datasets to maintain the neural network's capabilities.
>
> As for R2, we were not aware of this work since it has not yet been published in a peer-reviewed venue, but with only an arXiv version available. The authors of R2 propose a different method for machine unlearning, focusing on altering the entire model rather than selecting crucial weights. We will include this discussion in the camera-ready.
>
> In terms of comparison, we believe it is not feasible to compare SEMU with R2 due to its unpublished status and the unavailability of the code. However, we will include a comparison with R1 in our revised version.
>
> **Intuition behind SEMU** In practice, some directions are more important than others for all weights. Observe that the weights are roughly proportional to the averaged gradient over the entire dataset. However, if we consider only the subset (class) we want to unlearn, its gradient will share some directions with the gradient of the whole dataset, but will also have directions specific to that subset. Thus, the projection ensures that we remove the common directions from both the weights and the gradient of our subset. Consequently, during the unlearning process, we do not modify the directions crucial to the model but only those specific to the dataset.
>
> **Similarity to SalUn** We believe SalUn is the most similar to SEMU, as both methods aim to alter only crucial model weights based on gradient information. However, our approach significantly reduces the number of altered weights by up to 50 times. Additionally, SEMU uses the same loss functions when processing the forget dataset and can operate without a remaining dataset, addressing a limitation of SalUn. When SalUn operates only on the forgetting dataset, its performance diminishes as the model collapses (see response for Reviewer Qmc3).
>
> **Loss functions for SEMU** The training process and loss function of SEMU differ slightly, as SEMU does not rely on the remaining dataset. Therefore, the objective for the remaining set is not used in model optimization.
>
> **Metric definition** When it comes to metrics, we use the well-established ones in literature. When it comes to MIA, we use an MIA defined in [1], which was also used in SalUn and previous works.
>
> [1] Carlini, Nicholas, et al. "Membership inference attacks from first principles." 2022 SP.
>
> **Accuracy for generation task** To measure the accuracy of the image generation task, we use a classifier trained to recognize images generated by the model for a given class. We then apply this classifier to a newly generated batch of images after unlearning.
>
> **Time consumption** Please see the response for Reviewer Qmc3
>
> **MIA and UA worse than SalUn** We agree that the results presented in Tab. 6 and Tab. 7 are comparable or slightly worse than SalUn, especially in terms of the FID metric. This difference is mostly caused because of a different approach, i.e., we want to have an unlearning method working reasonably well, even without access to any examples from the remaining dataset, while SalUn utilizes the remaining dataset during unlearning.
>
> Finally, we observed that SalUn's concept of unlearning leads to catastrophic behaviour on normal (safe) prompts and conceptually far on NSFW as well. We provide the quantitative and qualitative results in Sections 2, 3, and 4 in additional experiments (https://anonymous.4open.science/r/icml2025_submission_3162/REBUTTAL.md).
>
> **Test set construction** The test set consists of test images from the dataset used for evaluation. In the case of random data forgetting, the test set remains unaltered. However, when performing class-wise forgetting, we remove the forgotten class from the test.
>
> **Forgetting labels** When forgetting, we perform random relabelling, meaning that we assign a random label from the remaining classes.
>
> **Additional benchmarks** See responses to Reviewers iRuu and Qmc3. We believe these experiments showcase SEMU's effectiveness and the results cover a broad range of task scales, capabilities, and complexities, while the proposed benchmarks are similar.
>
> **More samples for DDPM** Unlearning samples of other CIFAR10 classes generated with DDPM will be added to the revised version. Now, we present them at https://anonymous.4open.science/r/icml2025_submission_3162/REBUTTAL.md in Section 5.
>
> **Why remaining-free?** The dataset-free scenario can be justified for similar reasons as exemplar-free continual learning. Namely, due to privacy reasons (e.g. GDPR, CCPA), the remaining data may be inaccessible during the unlearning or too large to efficiently retrain on. Also, computational efficiency is improved by avoiding training episodes on the remaining data.

---

> > ### Comment · Reviewer_tPuz · 2025-04-07
> >
> > Thanks for the authors' response. However, there remain some unsettled issues.
> >
> > - Though there are no existing machine unlearning methods with available code, the authors should discuss the difference between LoRA and their method theoretically.
> >
> > - The authors say that they utilize trained models to recognize images generated by the model. Can the authors present the accuracy of each model to show the reliability of the recognition procedure?

---

> > > ### Author Response · Authors · 2025-04-08
> > >
> > > We thank the Reviewer for their response to our rebuttal.
> > >
> > > **On the classifier's accuracy**
> > >
> > > We followed the experimental scheme from the SalUn paper, and we used the pretrained classifiers for class generations with diffusion models. In particular, for DDPM and CIFAR10, we used Resnet34, achieving $94.97$\% accuracy. Whereas, for Stable Diffusion and Imagenette, we used Resnet50 (with the weights from torchvision), achieving acc@1=80.858\% and acc@5=95.434\% on ImageNet.
> > >
> > > **On the differences between SEMU and LoRA**
> > >
> > > While both LoRA and SVD involve low-rank matrices, they are fundamentally different in formulation and purpose. SVD identifies and surgically removes subspaces associated with specific data by editing or eliminating components correlated with that data. LoRA just learns an unconstrained additive update for adaptation. The low-rank matrix in SEMU is not merely a compression tool (as in, e.g., [1]), but a mechanism to enable interpretable and controlled unlearning.
> > >
> > > Moreover, as we already presented in the manuscript, SVD gives an optimal r-rank decomposition according to the singular values (see Theorem 1), whereas LoRA is a low-rank decomposition learned via gradient methods.
> > > SEMU uses SVD, because it gives us the orthonormal projections, resulting in the geometric separation of the features to unlearn from the remaining knowledge, being also easily interpretable. These properties are not true for the learnable projections, like LoRA.
> > >
> > > In the paper [2], authors decompose the weight matrix $W \in \mathbb{R}^{m \times n}$ into $AB + W^{res}$, where
> > >
> > > $A = U_{:r}S^{\frac{1}{2}}_{:r} \in \mathbb{R}^{m \times r}$
> > >
> > > and
> > >
> > > $B = S_{:r}^{\frac{1}{2}} V^{T}_{:r} \in \mathbb{R}^{r \times n}$.
> > >
> > > $A$ and $B$ correspond to $r$ principal singular values of $W$, and are further trained. $W^{res} = U_{r:}S_{r:}V^{T}_{r:} \in \mathbb{R}^{m \times n}$ is associated with residual singular values and remains frozen during fine-tuning. Such an approach surpasses LoRA in several experiments.
> > >
> > > Analogously, in the context of machine unlearning, SVD precisely selects the most important components related to the forget dataset and leaves the rest of the parameters intact.
> > >
> > > We sincerely appreciate your constructive comments and concerns, which help us improve our manuscript. We hope our detailed response effectively addresses your concerns. If so, we would appreciate it if you could increase the rating accordingly. Please feel free to ask if you have any additional questions.
> > >
> > > **References:**
> > >
> > > [1] Wang, X., Zheng, Y., Wan, Z., & Zhang, M. (2024). Svd-llm: Truncation-aware singular value decomposition for large language model compression. arXiv preprint arXiv:2403.07378.
> > >
> > > [2] Meng, F., Wang, Z., & Zhang, M. (2024). Pissa: Principal singular values and singular vectors adaptation of large language models. Advances in Neural Information Processing Systems, 37, 121038-121072.

---

### Official Review · Reviewer_8DF6 · 2025-03-15

**Overall Recommendation:** 2

**Summary:**

The paper proposes SEMU, a machine unlearning method using Singular Value Decomposition (SVD) to efficiently erase specific data influences from trained models. SEMU leverages SVD to project model gradients into a low-dimensional subspace, identifying critical weights linked to unwanted data. By updating a small ratio of parameters and eliminating reliance on original datasets, SEMU achieves competitive unlearning performance in image classification (CIFAR-10/100) and generation (DDPM, Stable Diffusion) tasks while preserving model utility.

**Claims And Evidence:**

The claims focus on two main topics: **competitive unlearning performance** and **improved efficiency**.

### **competitive unlearning performance**
Taking the results in the Appendix into account, the provided results for Random Data Forgetting seem to be good. However, the overall presentation is not clear, and the important metrics were not well explained, e.g., what's the difference between RA and TA, and why higher TA results were not in bold? In addition, this paper lacks of ablation studies for the proposed method.

### **improved efficiency**
> 'SEMU eliminates the dependency on the original training dataset (in abstract)'

SEMU achieves good performance on unlearning without $D_r$ (Tables 1–3). However, 'eliminates' may be too confident.

>  'SEMU minimizes the number of model parameters (in abstract)'

This is true when compared with the selected baselines. However, the evaluation with respect to time consumption is missing, which is important as the method involves extra computation cost to decide the trainable params.

**Essential References Not Discussed:**

Since SEMU changes the network structure by adding an extra component for each layer (Eq 13), a well-known similar method, namely LoRA, should be discussed.
Moreover, LoRA can be used for MU tasks, it may have less trainable parameters and better performance.

**Experimental Designs Or Analyses:**

Besides the issues mentioned above, additional issues:

- It's noted that TParams of SalUn is set to 50\% in Table 1, why set that to 100% in Table 2 and Table 3?
- The analysis in Table 5 and Figure 4 are for SalUn only, which contributes less to the evaluation of the proposed method. why not do the same for the proposed SEMU?
- The numeral evaluation for the image generation task (Table 6) is confusing, as UA seems to be a the-smaller-the-better metric in Table 1. And the TA has a huge gap compared to 'Retrain'. How to understand this result?

**Methods And Evaluation Criteria:**

The methods and evaluation criteria in the SEMU paper are largely appropriate for machine unlearning (MU) but have notable limitations.

Strengths:
- Theoretically grounded (Theorem 4.1) for low-rank approximation, aligning with MU’s goal of minimal parameter updates.
- Gradient projection logically preserves model performance.

Limitations:
- An important baseline is missing. Since SEMU changes the network structure, a similar method, namely LoRA (using LoRA for MU tasks), may have less trainable parameters and better performance.
- No analysis of SVD’s computational overhead.
- No sufficient ablation study was provided.

**Other Comments Or Suggestions:**

Typos:

- In the caption of Figure 2, "... adn(?) ...".
- In the second paragraph of Sec 3.2, "and often(?) $D_r$".

**Other Strengths And Weaknesses:**

See above.

**Questions For Authors:**

I have listed most of my questions above, additionally:

- Have you properly set the Latex template? It's expected that papers under review should have line numbers (using `\usepackage{icml2025}` instead of `\usepackage[accepted]{icml2025}`)

**Relation To Broader Scientific Literature:**

Yes. The SEMU paper positions its contributions within the broader machine unlearning (MU) literature by addressing two key limitations of prior work: parameter inefficiency and dependency on the remaining dataset ($D_r$).

**Theoretical Claims:**

There was a proof for Theorem 4.1, it correctly invokes the Eckart-Young-Mirsky theorem, which guarantees that truncated SVD minimizes the Frobenius norm error for rank-r approximations. The proof is too short that It's uncertain if any important assumptions were missing.

---

> ### Author Rebuttal · Authors · 2025-03-31
>
> **Time consumption**
> In the Table below we show a comparison of time needed to unlearn DDPM model:
> |Method|Preprocessing time|1000 iters time|
> |:-|:-:|:-:|
> |SEMU|44.18s|308s|
> |SEMU_retrain|44.18s|530s|
> |SalUn|50.69s|1170s|
>
> Also, we show the time of unlearning of ResNet-18:
> |Method|Dataset|Preprocessing time|One unlearning epoch|
> |:-|:-:|:-:|:-:|
> |SEMU|CIFAR-10|3.27s|11.01s|
> |SalUn|CIFAR-10|2.25s|14.70s|
> |SEMU|CIFAR-100|3.36s|11.31s|
> |SalUn|CIFAR-100|2.20s|14.85s|
>
> **Bolded values in the tables** highlight two key metrics: target accuracy, reflecting the similarity between a model’s performance and its retrained version, and the number of parameters modified during unlearning. This emphasizes that SEMU minimally affects model behavior compared to other methods. For generative models, we evaluate stable diffusion outputs before and after unlearning using a safe prompt, identical seed, and noise. Notably, we observe that SalUn exhibits drastic changes in model behavior (not ability), which are not evident with SEMU.
>
> **Ablations** Here we provide ablations on the projection usage in SEMU:
>
> |Dataset|Task|Projection|UA|RA|TA|MIA|
> |:-|:-:|:-:|:-:|:-:|:-:|:-:|
> |CIFAR10|Random 10%|No|3.80(1.44)|96.46(3.54)|89.78(4.48)|11.64(1.24)|
> |CIFAR10|Random 10%|Yes|0.60(4.64)|99.40(0.60)|94.22(0.04)|5.40(7.48)|
> |CIFAR10|Random 50%|No|2.13(5.78)|97.69(2.31)|91.17(0.55)|8.37(10.92)|
> |CIFAR10|Random 50%|Yes|1.77(6.14)|98.12(1.88)|91.80(0.08)|7.20(12.09)|
> |CIFAR10|Class Forget.|No|99.72(0.28)|98.55(1.45)|92.65(0.60)|100.00(0.00)|
> |CIFAR10|Class Forget.|Yes|99.83(0.17)|98.22(1.78)|92.26(0.21)|100.00(0.00)|
>
> One can observe that the projection positively influences the unlearning process by preserving model capabilities what can be seen in RA and TA metrics, while slightly widening the gap between retrain model and unlearned one in FA and MIA.
>
> Another ablation on the parameter choosing the portion of variance explained by the SVD that is used for selecting parameters for alteration is in the rebuttal for Reviewer iRuu.
>
> **Comparison with LoRA** Thank you for your feedback. We are open to comparing SEMU with the LoRA method for machine unlearning. Could you please direct us to a specific LoRA-based machine unlearning method with available code? This will enable us to conduct a fair comparison. If such a method does not exist, we believe that adapting LoRA for machine unlearning is beyond the scope of our current work, as it would require significant effort.
>
> **Theoretical claims** Thank you for noticing the concern with the proof. We know that the Eckart-Young-Mirsky theorem (matrix approximation lemma) is true for any unitarily invariant norm. In particular, it is true for the Frobenius norm too. Throughout the paper, while introducing SVD formally, we assume the Frobenius norm and operate in a Hilbert space. Since each element of $S^r$ is of rank $r$ at most, we know that among all $r$-rank matrix approximations for $G$, the optimal one is the one given by the SVD. Moreover, we know that the optimal solution is unique. In the revised version of this manuscript, we will add the Eckart-Young-Mirsky theorem and all the needed assumptions in the same place, in order to make the proof more approachable.
>
> **100% of TParams for SalUn in Supplement** We are grateful to the Reviewer for catching the typo. It was an oversight due to a copy-paste error. We apologize for the mistake. In the camera-ready version of the paper, we will correct the values in Tables 2 and 3 for SalUn (50% instead of 100%).
>
> **Table 4 and Table 5 are only for SalUn** In our evaluation of SalUn, we aimed to demonstrate that when fewer parameters are altered than originally described, SalUn's effectiveness diminishes, regardless of the data. To achieve a fair comparison, we conducted an experiment using the same setup but altering only 1% of the weights with SalUn. The results can be found in a rebuttal for Reviewer Qmc3.
>
> **Numerical evaluation for the generation task**
> We thank the Reviewer for a question regarding their concern. We perform a similar analysis of the generative diffusion models behavior as for classification tasks. That’s the rationale behind reporting TA as well. Following the SalUn evaluation procedure, firstly, we pretrained the same classifier to analyse the generated samples. We observed that even a small change in generated features is hard for such a classifier. Following the Reviewer's concern, we want to admit that the TA metric for unlearning in the generative models scenario is a biased metric. For the next comparisons (e.g., the ones from Section 1 in https://anonymous.4open.science/r/icml2025_submission_3162/REBUTTAL.md), we focused on FID and UA metrics.
>
> **Template usage** Thank you very much for spotting this issue. We are very sorry for the mistake and any difficulties this could cause.
>
> **Typos** We would like to thank the Reviewers for their thorough work in helping us improve our manuscript. We apologize for typos.

---

### Official Review · Reviewer_Dn3F · 2025-03-18

**Overall Recommendation:** 4

**Summary:**

The authors proposed a Singular Value Decomposition for Efficient Machine Unlearning method which solve two problems 1) the need remaining dataset for unlearning process and 2) changes too many parameters during unlearning process

**Claims And Evidence:**

This article show good evidence to support its claim that their SEMU method show reasonable performance (superior or matching) to existing method while simpler in sense of optimization

**Essential References Not Discussed:**

NA

**Experimental Designs Or Analyses:**

I think the experimental part is the weakest point of this paper. The author only shows results of image classfication and image generation results. Over task such as NLP and math are needed to show how versatile this unlearning method is

**Methods And Evaluation Criteria:**

The nature of the method , in layman language, is similar to group parameters (gradient matrix) into r cluster (where r is dimension of the truncated SVD). Overall it makes a lot of sense. The math reasoning and results also support it.

**Other Comments Or Suggestions:**

NA

**Other Strengths And Weaknesses:**

NA

**Questions For Authors:**

The most important questions is why choose SVD to select parameters?

**Relation To Broader Scientific Literature:**

unlearning can be used in many fields of sciences and beyong

**Theoretical Claims:**

The theory part of this study is simple and straight forward based on basic SVD and based on previous paper in the literature

---

> ### Author Rebuttal · Authors · 2025-03-31
>
> **Why SVD?** Our objective was to minimize the number of altered weights during the unlearning process to maintain the model's behavior. To achieve this, we looked for an effective selection mechanism. SVD, our first choice, proved to be successful, so we did not explore other parameter selection methods. However, investigating alternative selection methods for potential improvements in effectiveness and efficiency is an interesting path for future work.
>
> **Experiments on TinyImageNet with ViT and ResNet** To futher show applicability of SEMU, we provide results on TinyImageNet for ResNet18 and ViT models.
>
> Performance on ResNet-18, pre-trained on Tiny ImageNet dataset, for 10% random data forgetting.
> |Methods|UA|RA|TA|MIA|
> |:-|:-:|:-:|:-:|:-:|
> |Retrain|36.40|99.98|63.67|63.77|
> |ℓ1-sparse|15.19(21.21) |98.61(1.37)|61.78(1.89)|26.39(37.38)|
> |SalUn|27.78(8.62)|97.20(2.78)|59.70(3.97)|72.80(9.03)|
> |SEMU|5.44(30.96)|95.02(4.96)|64.03(0.36)|15.18(48.59)|
> |SEMU_remain|5.08(31.32)|94.98(5.00)|63.77(0.10)|20.81(42.96)|
>
> Performance on ResNet-18, pre-trained on Tiny ImageNet dataset, for 1 random class (number 9) data forgetting.
> |Methods|UA|RA|TA|MIA|
> |:-|:-:|:-:|:-:|:-:|
> |Retrain|100.00|99.98|64.21|100.00|
> |ℓ1-sparse|44.00(56.00)|62.76(37.22)|49.93(14.28)|50.60(49.40)|
> |SEMU|20.80(79.20)|95.52(4.46)|64.95(0.74)|44.60(55.40)|
> |SEMU_remain|65.80(34.20)|96.17(3.81)|64.67(0.46)|87.80(12.20)|
>
> Performance on ViT, pre-trained on Tiny ImageNet dataset, for 10% random data forgetting.
> |Methods|UA|RA|TA|MIA|
> |:-|:-:|:-:|:-:|:-:|
> |Retrain|14.30|99.91|85.59|24.61|
> |SEMU|2.14 (12.16)|95.01 (4.90)|85.85 (0.26)|5.87 (18.74)|
> |SEMU_remain|2.00 (12.30)|94.96 (4.95)|85.48 (0.11)|8.04 (16.57)|
>
> Performance on ViT, pre-trained on Tiny ImageNet dataset, for 1 random class (number 9) data forgetting.
> |Methods|UA|RA|TA|MIA|
> |:-|:-:|:-:|:-:|:-:|
> |Retrain|100.0|99.91|85.37|100.0|
> |SEMU|20.80 (79.20)|95.47 (4.44)|84.09 (1.28)|44.50 (55.50)|
> |SEMU_remain|65.80 (34.20)|96.12 (3.79)|85.10 (0.27)|87.70 (12.30)|

---

### Official Review · Reviewer_iRuu · 2025-03-20

**Overall Recommendation:** 3

**Summary:**

The paper "SEMU: Singular Value Decomposition for Efficient Machine Unlearning" introduces a new method for machine unlearning (MU). The goal is to remove specific data from AI models without damaging overall performance. Traditional unlearning methods require modifying large portions of the model or retraining with remaining data. This makes them computationally expensive and impractical for privacy-sensitive applications.

SEMU solves these issues by using Singular Value Decomposition (SVD). Instead of altering the entire model, SEMU identifies and modifies only the most crucial weights linked to the data that needs to be forgotten. This makes the process faster and more efficient, with minimal impact on the model’s generalization ability.

The paper demonstrates SEMU’s effectiveness through experiments on image classification (CIFAR-10, CIFAR-100) and image generation (Stable Diffusion, DDPMs). The results show that SEMU can achieve strong unlearning performance while modifying less than 1% of the model’s parameters. It also works without requiring access to the original training dataset, making it ideal for privacy-focused applications.
In conclusion, SEMU provides an efficient, data-independent, and computationally lightweight approach to machine unlearning. It outperforms existing methods in efficiency while maintaining accuracy. The authors suggest that SEMU could be extended to large language models (LLMs) and vision-language models (VLMs) in future research.

**Claims And Evidence:**

The paper makes several key claims about SEMU's effectiveness, efficiency, and practicality in machine unlearning. The main claims are:

1.	SEMU achieves efficient unlearning by modifying only a small fraction of model weights (~1%) instead of retraining the entire model.
2.	SEMU does not require access to the remaining dataset, making it more privacy-friendly than traditional methods.
3.	SEMU maintains model accuracy while effectively removing unwanted knowledge.
4.	SEMU outperforms other unlearning methods in both image classification and image generation tasks.

These claims are backed by extensive experimental results on CIFAR-10, CIFAR-100, and Stable Diffusion models. The paper provides detailed comparisons against existing methods like SalUn, ESD, and Forget-Me-Not (FMN). The results show that SEMU achieves similar or better unlearning performance while altering far fewer model parameters.

The claim that SEMU eliminates the need for the remaining dataset is well-supported. The experiments show that even without access to retained data, SEMU still performs effective unlearning with minimal accuracy loss. However, the paper does acknowledge that having access to some remaining data can further improve results.

There are no major unsupported claims in the paper. The methodology, theoretical background, and experiments provide clear and convincing evidence to validate SEMU’s advantages. The only area where further research may be needed is in applying SEMU to different architectures like large language models (LLMs).

**Essential References Not Discussed:**

The paper presents a strong foundation by referencing key works in machine unlearning, model compression, and SVD-based optimizations. However, some critical prior research is missing, which could strengthen the context of SEMU’s contributions.
For example, the paper introduces a selective unlearning method using SVD, arguing that modifying only a small fraction of model parameters (~1%) is sufficient.

However, prior works on low-rank decomposition in deep learning have studied similar principles in different contexts but are not cited here.
Example of a missing reference:
"The key contribution of this paper is an efficient machine unlearning method using low-rank SVD, modifying fewer parameters than prior approaches. However, previous work by Denton et al. (2014) proposed a low-rank decomposition method for CNN compression, which also showed that selective weight modification can preserve model performance. While SEMU applies this concept to unlearning, acknowledging this prior work would provide stronger theoretical grounding."

Similarly, SEMU claims that unlearning can be achieved without access to the remaining dataset. However, studies like Wu et al. (2022), "PUMA: Provable Machine Unlearning", provide mathematically provable guarantees for unlearning but are not cited. Including this reference would help differentiate SEMU’s empirical approach from provable unlearning techniques.
By adding references to low-rank model adaptation, provable machine unlearning, and privacy-preserving ML, the paper could provide a more comprehensive context for its

**Experimental Designs Or Analyses:**

The experimental design in this paper is well-structured and rigorous. The authors carefully design tests for both image classification and image generation tasks to evaluate SEMU’s unlearning performance. They also compare SEMU against multiple baseline methods, ensuring a fair and meaningful comparison.

**Methods And Evaluation Criteria:**

The paper uses logical and well-structured methods to evaluate SEMU. The authors test their approach on both classification and generative models, ensuring broad applicability. They compare SEMU against state-of-the-art machine unlearning methods, including SalUn, ESD, and FMN, using widely accepted benchmarks.

**Other Comments Or Suggestions:**

NONE

**Other Strengths And Weaknesses:**

Strengths
1.	The use of Singular Value Decomposition (SVD) for selective forgetting is a novel contribution to the machine unlearning field. Unlike prior methods that require full model retraining or large-scale fine-tuning, SEMU modifies only a small fraction of model weights (~1%), making it computationally efficient.
2.	One of SEMU’s most significant advantages is its ability to perform unlearning without access to the remaining dataset. This is a major step forward for privacy-preserving AI, where retraining with retained data is often impractical.
3.	The authors conduct extensive experiments on both classification and generative models, covering datasets like CIFAR-10, CIFAR-100, and Stable Diffusion. The results are compared against state-of-the-art MU methods (SalUn, ESD, Forget-Me-Not), demonstrating SEMU’s superior efficiency and effectiveness.

Weaknesses
1.	SEMU is an empirical approach, meaning it lacks formal mathematical guarantees for unlearning effectiveness. Prior work, such as PUMA (Wu et al., 2022), provides provable unlearning methods, whereas SEMU relies on experimental validation rather than formal proofs.
2.	While SEMU is tested on image classification and generative models, it is not evaluated on large-scale architectures like transformers or LLMs. Applying SEMU to text-based models (e.g., BERT, GPT-4, or ViTs) would strengthen its generalizability.
3.	The paper does not fully explore how different SVD truncation levels affect unlearning performance. An ablation study on how much of the singular value spectrum needs modification could help optimize SEMU’s implementation further.
4.	While SEMU is described as computationally efficient, GPU/memory usage details for different architectures are not fully reported. A detailed breakdown of training costs compared to full retraining methods would provide more clarity on real-world feasibility.

**Questions For Authors:**

1.	How does SEMU perform on large-scale architectures such as transformers and large language models (LLMs)?
2.	How sensitive is SEMU’s performance to different levels of SVD truncation?
3.	What specific hyperparameters influence SEMU’s efficiency the most?

**Relation To Broader Scientific Literature:**

The paper builds on prior research in machine unlearning, matrix factorization, and model compression, integrating these concepts into a novel approach. SEMU’s use of Singular Value Decomposition (SVD) for unlearning connects to several existing fields in AI and machine learning.

**Theoretical Claims:**

The paper presents a theoretical foundation for SEMU, primarily based on Singular Value Decomposition (SVD) and its ability to reduce model parameters in a structured way. The theoretical claims focus on why SEMU is effective for machine unlearning and how modifying a small subset of model weights can achieve efficient forgetting without damaging overall performance.

---

> ### Author Rebuttal · Authors · 2025-03-31
>
> Thank you for your thorough review. We would like to address some of your concerns below:
>
> **Time needed for SEMU when compared to SalUn** In the Table below we show a comparison of time needed to unlearn DDPM model:
> |Method|Preprocessing time|1000 iters time|
> |:-|:-:|:-:|
> |SEMU|44.18s|308s|
> |SEMU_retrain|44.18s|530s|
> |SalUn|50.69s|1170s|
>
> Also, we show the time of SEMU and SalUn in unlearning of CIFAR10 and CIFAR100 for ResNet-18:
> |Method|Dataset|Preprocessing time|One unlearning epoch|
> |:-|:-:|:-:|:-:|
> |SEMU|CIFAR-10|3.27s|11.01s|
> |SalUn|CIFAR-10|2.25s|14.70s|
> |SEMU|CIFAR-100|3.36s|11.31s|
> |SalUn|CIFAR-100|2.20s|14.85s|
>
> When it comes to memory usage, SEMU does not require additional storage, so it requires the same amount of memory as SalUn, or slightly less, as we do not require a mask of neurons which needs to be altered.
>
> **SEMU for large architectures** In this experiment, we demonstrate the effectiveness of SEMU using the TinyImageNet dataset and the ViT model. We tested SEMU's ability to forget 10% of randomly chosen data and one entire class. The results indicate that SEMU maintains strong performance with the ViT model in both scenarios.
> Regarding the application of SEMU to Large Language Models (LLMs), it's important to note that our focus has been on models designed for computer vision. Therefore, adapting SEMU to LLMs may be beyond the scope of this work.
>
> |Methods|UA|RA|TA|MIA|
> |:-|:-:|:-:|:-:|:-:|
> |Retrain|14.30|99.91|85.59|24.61|
> |SEMU|2.14 (12.16)|95.01 (4.90)|85.85 (0.26)|5.87 (18.74)|
> |SEMU_remain|2.00 (12.30)|94.96 (4.95)|85.48 (0.11)|8.04 (16.57)|
>
> |Methods|UA|RA|TA|MIA|
> |:-|:-:|:-:|:-:|:-:|
> |Retrain|100.0|99.91|85.37|100.0|
> |SEMU|20.80 (79.20)|95.47 (4.44)|84.09 (1.28)|44.50 (55.50)|
> |SEMU_remain|65.80 (34.20)|96.12 (3.79)|85.10 (0.27)|87.70 (12.30)|
>
> **Referring to other low-rank adaptation methods**. We will update the discussion on low-rank adaptation methods in the final version of our work, which will include PUMA.
>
> **Different SVD truncation levels influence on model's performance** Parameter $r$ gives the size of a submatrix of the SVD projection, which we use for unlearning purposes. In particular, in each changed layer $L$, we are setting the value of $r_L$ to be the same percentage alpha of the rank of this matrix.
>
> This procedure gives us the square submatrices of sizes $r_L \times r_L$. Please note that the value of $r_L$ is different for various layers, however, in each layer, we have the same percentage (alpha) of important directions in the SVD projection. We consider SEMU_retrain (Tab. 1) and SEMU_subset (Tab. 2) scenarios.
>
> Tab. 1
>
> |alpha|UA|FID|
> |:-|:-:|:-:|
> |0.01|100.00|16.64|
> |0.05|100.00|17.83|
> |0.1|100.00|17.83|
> |0.2|100.00|17.36|
> |0.3|100.00|17.39|
> |0.4|100.00|17.40|
> |0.5|100.00|17.39|
>
> Tab. 2
>
> |alpha|UA|FID|
> |:-|:-:|:-:|
> |0.01|100.00|17.17|
> |0.05|98.00|18.20|
> |0.1|100.00|17.83|
> |0.2|100.00|17.74|
> |0.3|100.00|17.72|
> |0.4|100.00|17.72|
> |0.5|100.00|17.57|
>
> We observe that the proposed method of selecting the most important directions is more efficient than selecting just a percentage of the low-rank projection matrix (better UA and FID). For naive selection, we observe that the best results are for alpha=0.01. Then, the metric values are higher, but not linearly.
>
> **Most important parameters for SEMU**
> The effectiveness of SEMU is influenced by several factors, due to the fine-tuning of crucial model parameters. From the standpoint of the machine unlearning method, the parameter $\gamma$ is important. It is responsible for the selection of weights that are modified during the unlearning process. Specifically, $\gamma$ selects weights with an SVD variance no less than its value (e.g. 90% of variance), enabling the identification of a critical subset of weights for alteration. For further details, please refer to the SEMU section and Eq. 12.

---

### Official Review · Reviewer_Qmc3 · 2025-03-25

**Overall Recommendation:** 3

**Summary:**

The paper performs an SVD decomposition for machine unlearning, which enables them for efficient unlearning. They also propose a dataset-free scenario, addressing data privacy concerns. Experiments show their superiority over other methods


At the core, SEMU aims to change a minimal number of model parameters, with a goal of removal of unwanted knowledge. They pose this problem as minimizing d(G; S^{r} _{A,B}) where G is the underlying loss function and S is subspace matrix induced by the matrices A and B.
The paper finally performs experiments for class unlearning and image generation.

**Claims And Evidence:**

The paper claims several points:


**The paper claims that their method is theoretically substantiated in the sentence quoted as follows:**


 "To overcome the challenges of gradient-based unlearning, we propose a novel, theoretically grounded method for selecting the most significant subspace of weights, θs , derived from the forgetting dataset Df ."
This claim is not substantiated. One possibility to show some generalization bound. I was wondering if the authors could connect to stability (Olivier Bosquet et al., Stability and Generalization, 2002) and then use them to show some generalization results. Note that Bosquet et al. showed stability in the context of perturbation of “data” rather than model perturbation. Specifically, they showed that in the presence of a regularizer, the weight values w satisfy:
$$ ||w(D)-w(D’)|| = O(1/|D|) \text{ where }  |D\backslash D’|=1 $$


This result is further used to show a generalization guarantee.  This paper is a great recipe for understanding the above problem's " dual ": Can the performance be stable and therefore enjoy a better generalization guarantee if we minimize Eq. (10), which directly ensures the S is close to G?


Given the short rebuttal timeframe, it is absolutely OK not to go for a complete proof of theory, but some discussion on connection would be helpful. Otherwise, removal of the above sentence may be better.




**The paper claims to perform efficient unlearing.**


However, I did not find (or perhaps did not understand) how the model is efficient. If the unlearning method is efficient, then it would be good to obtain a tradeoff plot between accuracy and efficiency. This is particularly important, because for example, a simple randomization can be very efficient but inaccurate and in this paper, SVD may consume time.


**Extensive experimental validation**


Since the paper claims to perform efficient unlearning, it would be important to perform experiments using Imagenet dataset or tinyImagenet dataset. To the best of my understanding, the paper performs image generation using a subset of Imagenet dataset. However, they are not usign Imagenet for classification task. CIFAR10 or CIFAR100 may be less challenging for this task. I understand that it may be difficult to perform experiments on imagenet during rebuttal period. Could the authors perform experiments using tinyimagenet instead for class prediction?

**Essential References Not Discussed:**

I do not see any such obvious reference missed.

**Experimental Designs Or Analyses:**

As I mentioned in the claims and evidence, experiments with large datasets are important in this context. Tinyimagenet can be a good candidate. Moreover, a tradeoff plot between the time and accuracy plus time and memory can be helpful.


Also, it would great if the authors can compare or discuss the conenction with various data subset selection methods, including Pruning, RHOLoss, GradMatch etc., which performs training on a small subset of data. Specifically, can we select the forget batch using one of these methods?

**Methods And Evaluation Criteria:**

The evaluation/experiments can be divided into two clusters: One is quantitative, and the other is qualitative. Re. For quantitative experiments, which are more metric based, I would prefer a trade-off between accuracy and time (both training and inference) and unlearning accuracy and time (both training and inference). The authors can also compare accuracy and memory (say how much GPU memory being consumed). Is SEMU pareto optimal in those curves?


There is not much ablation study of different components of their approach. For example, what is the benefit of Projection gradient improvement? It is not clearly understood.


Qualitative experiments are OK.

**Other Comments Or Suggestions:**

Minor: In introduction: Our contributions "ca be" summarized as follows ---> Our contributions "can be" summarized as follows:

**Other Strengths And Weaknesses:**

Apart from my points, I think the paper needs some reorganization. For example, many important details— from algorithms to loss functions are deferred to Appendix. Eq. (17,18) can be brought back to main.

**Questions For Authors:**

1. Can the authors show some theoretical insights which can shed light on generalization bounds?


2. How will the approach perform for large datasets?


3. How well this approach trade off in terms of time and memory? Can we have a trade off curve?

================

After rebuttal: Authors sufficiently addressed my concerns. But after looking at other reviews, I decided to keep my score.

**Relation To Broader Scientific Literature:**

The authors did a great job in the related work. But more papers and conenction with differential privacy and data subset selection would be better.

**Theoretical Claims:**

As I mentioned in the claims and evidence, the theoretical justification is not adequate. Can the authors leverage some existing results of stability to show some generalization bound?

---

> ### Author Rebuttal · Authors · 2025-03-31
>
> To address the concerns and questions raised by the Reviewer, we would like to point out the following:
>
> **More on theoretical aspects of projection**. In practice, some directions are more important than others for all weights. Observe that the weights are roughly proportional to the averaged gradient over the entire dataset. However, if we consider only the subset (class) we want to unlearn, its gradient will share some directions with the gradient of the whole dataset, but will also have directions specific to that subset. Thus, the projection ensures that we remove the common directions from both the weights and the gradient of our subset. Consequently, during the unlearning process, we do not modify the directions crucial to the model but only those specific to the dataset. We thank the Reviewer for a related paper. We will work more on theoretical guarantees.
>
> **Performance on larger dataset**. We have run experiments on TinyImageNet and here are the results:
>
> Performance on ResNet-18, pre-trained on Tiny ImageNet dataset, for 10% random data forgetting.
> |Methods|UA|RA|TA|MIA|
> |:-|:-:|:-:|:-:|:-:|
> |Retrain|36.40|99.98|63.67|63.77|
> |ℓ1-sparse|15.19(21.21) |98.61(1.37)|61.78(1.89)|26.39(37.38)|
> |SalUn|27.78(8.62)|97.20(2.78)|59.70(3.97)|72.80(9.03)|
> |SEMU|5.44(30.96)|95.02(4.96)|64.03(0.36)|15.18(48.59)|
> |SEMU_remain|5.08(31.32)|94.98(5.00)|63.77(0.10)|20.81(42.96)|
>
> Performance on ResNet-18, pre-trained on Tiny ImageNet dataset, for 1 random class (number 9) data forgetting.
> |Methods|UA|RA|TA|MIA|
> |:-|:-:|:-:|:-:|:-:|
> |Retrain|100.00|99.98|64.21|100.00|
> |ℓ1-sparse|44.00(56.00)|62.76(37.22)|49.93(14.28)|50.60(49.40)|
> |SEMU|20.80(79.20)|95.52(4.46)|64.95(0.74)|44.60(55.40)|
> |SEMU_remain|65.80(34.20)|96.17(3.81)|64.67(0.46)|87.80(12.20)|
>
> One can observe that in both experiments, SEMU achieves the lowest gap in target accuracy of the model.
>
> **Time needed for SEMU**
> Comparison of the time needed for SEMU and SalUn for the DDPM model. As can be observed, SEMU requires much less time to perform unlearning.
> |Method|Preprocessing time|1000 iters time|
> |:-|:-:|:-:|
> |SEMU|44.18s|308s|
> |SEMU_retrain|44.18s|530s|
> |SalUn|50.69s|1170s|
>
> Here are also the results for unlearning for the ResNet18 model:
> |Method|Dataset|Preprocessing time|One unlearning epoch|
> |:-|:-:|:-:|:-:|
> |SEMU|CIFAR-10|3.27s|11.01s|
> |SalUn|CIFAR-10|2.25s|14.70s|
> |SEMU|CIFAR-100|3.36s|11.31s|
> |SalUn|CIFAR-100|2.20s|14.85s|
>
> When it comes to memory usage, SEMU does not require additional storage, so it requires the same amount of memory as SalUn, or slightly less, as we do not require a mask of neurons which needs to be altered.
>
> **Comparison to SalUn within similar experimental conditions** Additionally, we ran experiments with SalUn, showcasing that SalUn collapses when the remaining dataset is not used during the unlearning procedure, no matter the amount of parameters altered (1% and 100%).
>
> |Method|Dataset|Task|With Remain Data|RA|TA|UA|MIA|TParams|
> |:-|:-:|:-:|:-:|:-:|:-:|:-:|:-:|:-:|
> |SalUn|CIFAR10|Random 10%|Yes|99.52|93.66|0.82|6.38|1%|
> |SalUn|CIFAR10|Random 10%|No|12.86|12.47|86.91|67.76|1%|
> |SalUn|CIFAR10|Random 10%|Yes|98.03|92.41|5.51|16.38|100%|
> |SalUn|CIFAR10|Random 10%|No|18.70|18.21|81.93|3.18|100%|
> |SEMU|CIFAR10|Random 10%|No|99.40|94.22|0.60|5.40|0.54%|
>
> |Method|Dataset|Task|With Remain Data|RA|TA|UA|MIA|TParams|
> |:-|:-:|:-:|:-:|:-:|:-:|:-:|:-:|:-:|
> |SalUn|CIFAR10|Class Forget.|Yes|99.65|94.83|93.35|100.00|1%|
> |SalUn|CIFAR10|Class Forget.|No|32.23|31.44|87.79|89.51|1%|
> |SalUn|CIFAR10|Class Forget.|Yes|99.48|93.94|99.99|100.00|100%|
> |SalUn|CIFAR10|Class Forget.|No|13.88|13.65|76.64|41.48|100%|
> |SEMU|CIFAR10|Class Forget.|No|98.22|92.26|99.83|100.00|0.87%|
>
> |Method|Dataset|Task|With Remain Data|RA|TA|UA|MIA|TParams|
> |:-|:-:|:-:|:-:|:-:|:-:|:-:|:-:|:-:|
> |SalUn|CIFAR100|Random 10%|Yes|97.46|72.74|4.13|22.44|1%|
> |SalUn|CIFAR100|Random 10%|No|1.44|1.29|98.56|0.71|1%|
> |SalUn|CIFAR100|Random 10%|Yes|98.83|67.17|64.69|91.76|100%|
> |SalUn|CIFAR100|Random 10%|No|0.97|1.14|98.84|9.96|100%|
> |SEMU|CIFAR100|Random 10%|No|97.39|74.14|2.53|8.82|1.18%|
>
> **Influence of projection on results**
>
> |Dataset|Task|Projection|UA|RA|TA|MIA|
> |:-|:-:|:-:|:-:|:-:|:-:|:-:|
> |CIFAR10|Random 10%|No|3.80(1.44)|96.46(3.54)|89.78(4.48)|11.64(1.24)|
> |CIFAR10|Random 10%|Yes|0.60(4.64)|99.40(0.60)|94.22(0.04)|5.40(7.48)|
> |CIFAR10|Random 50%|No|2.13(5.78)|97.69(2.31)|91.17(0.55)|8.37(10.92)|
> |CIFAR10|Random 50%|Yes|1.77(6.14)|98.12(1.88)|91.80(0.08)|7.20(12.09)|
> |CIFAR10|Class Forget.|No|99.72(0.28)|98.55(1.45)|92.65(0.60)|100.00(0.00)|
> |CIFAR10|Class Forget.|Yes|99.83(0.17)|98.22(1.78)|92.26(0.21)|100.00(0.00)|
>
> One can observe that the projection positively influences the unlearning process by preserving model capabilities, which can be seen in RA and TA metrics, while slightly widening the gap between retrain model and the unlearned one in FA and MIA.
>
> We are grateful for the review, and we are looking forward to a fruitful discussion on the provided answers.

---

### Decision · Program_Chairs · 2025-05-01

**Decision:**

Accept (poster)

**Comment:**

This paper proposes a method  for selective forgetting of specific data points from a pretrained ML model. They use the classical  Singular Value Decomposition idea to reduce the requirement for finetuning of large number of parameters.

This paper received 6 reviews out of which 4 have recommended acceptance, post rebuttal. I feel that the comments from the reject reviews can be addressed in the final revision and therefore I suggest acceptance.